# Soluble ST2 as a Potential Biomarker for Risk Assessment of Pulmonary Hypertension in Patients Undergoing TAVR?

**DOI:** 10.3390/life12030389

**Published:** 2022-03-08

**Authors:** Elke Boxhammer, Moritz Mirna, Laura Bäz, Nina Bacher, Albert Topf, Brigitte Sipos, Marcus Franz, Daniel Kretzschmar, Uta C. Hoppe, Alexander Lauten, Michael Lichtenauer

**Affiliations:** 1Division of Cardiology, Department of Internal Medicine II, Paracelsus Medical University of Salzburg, 5020 Salzburg, Austria; e.boxhammer@salk.at (E.B.); m.mirna@salk.at (M.M.); n.bacher@salk.at (N.B.); a.topf@salk.at (A.T.); b.sipos@salk.at (B.S.); u.hoppe@salk.at (U.C.H.); 2Department of Cardiology, Clinic of Internal Medicine I, Friedrich Schiller University Jena, 07743 Jena, Germany; laura.baez@med.uni-jena.de (L.B.); marcus.franz@med.uni-jena.de (M.F.); daniel.kretzschmar@med.uni-jena.de (D.K.); 3Department of General and Interventional Cardiology and Rhythmologoy, Helios Hospital Erfurt, 99089 Erfurt, Germany; alexander.lauten@helios-gesundheit.de; 4Deutsches Zentrum für Herz-Kreislauf-Forschung (DZHK), Standort Berlin, 10785 Berlin, Germany

**Keywords:** aortic valve stenosis, biomarker, right heart catheterization, soluble suppression of tumorigenicity-2, transcatheter aortic valve replacement

## Abstract

Background: Severe aortic valve stenosis (AS) is associated with pulmonary hypertension (PH) and has been shown to limit patient survival. Soluble suppression of tumorigenicity-2 (sST2) is a cardiovascular biomarker that has proven to be an important prognostic marker for survival in patients undergoing transcatheter aortic valve replacement (TAVR). The aim of this study was to assess the importance of the sST2 biomarker for risk stratification in patients with severe AS in presence or absence of PH. Methods: In 260 patients with severe AS undergoing TAVR procedure, sST2 serum level concentrations were analyzed. Right heart catheter measurements were performed in 152 patients, with no PH detection in 43 patients and with PH detection in 109 patients. Correlation analyses according to Spearman, AUROC analyses and Kaplan–Meier curves were calculated. Results: Patients with severe AS and PH showed significantly higher serum sST2 concentrations (*p* = 0.006). The sST2 cut-off value for non-PH patients regarding 1-year survival yielded 5521.15 pg/mL, whereas the cut-off value of PH patients was at a considerably higher level of 10,268.78 pg/mL. A cut-off value of 6990.12 pg/mL was related with a significant probability of PH presence. Survival curves showed that patients with severe AS and PH not only had higher 1-year mortality, but also that increased levels of sST2 plasma concentration were associated with earlier death. Conclusion: sST2 definitely has the potential to provide information about the presence of PH in patients with severe AS, in a noninvasive way.

## 1. Introduction

After mitral valve regurgitation, AS constitutes the second most frequent valvular disorder worldwide, but represents the most common disorder requiring medical treatment, with a prevalence of 3.4% in patients >75 years [1].

In addition to surgical aortic valve replacement, TAVR has been available since 2002 as a minimally invasive procedure that allows high-risk patients with severe pre-existing comorbidities, high frailty score, and limited anesthetic capacity to undergo a procedure that is now very safe and less risky [2]. In a population of patients >80 years of age, TAVR not only leads to a very satisfactory survival rate [3], but also to an improvement in quality of life of these patients [4].

Previous studies have repeatedly described that PH presented in addition to AS limits the prognosis of survival [5,6,7,8]. To verify the diagnosis of PH according to current European Society for Cardiology guidelines [9], right heart catheterization is the absolute gold standard. The information obtained about mean pulmonary arterial pressure (mPAP) and pulmonary arterial wedge pressure (PAWP) allows us to decide on the presence or absence of PH and, in case of presence, to further differentiate between pre- and post-capillary PH.

The suppression of tumorigenicity-2 (ST2) belongs to the Toll-like/Interleukin-1 receptor family and exists in the form of two isoforms—a membrane-bound (ST2 receptor or ST2L) and a soluble variant (sST2) [10]. In addition to cells of the specific immune defense, this protein is also expressed by cardiomyocytes as well as cardiac fibroblasts and endothelial cells. Mechanical stress or stretching of the myocardium by cardiac disease of any kind leads to increased release of interleukin-33 (IL-33) from fibroblasts, mast cells, endothelial cells and epithelial cells [11,12]. IL-33 binds as a ligand to the ST2 receptor and thereby initiates a signaling cascade with cardioprotective effects to prevent cardiac remodeling and consequent heart failure [13]. With sST2 as a so-called decoy receptor, a direct antagonist to the ST2 receptor is available, as sST2 binds IL-33 with high affinity, but does not lead to any signaling. Thus, less IL-33 is available as a ligand for ST2L, which is why the cardioprotective effect is at least partially eliminated [14]. This competitive inhibition makes it clear that in the presence of high sST2 concentrations, heart is exposed to greater stress reactions. It has already been successfully demonstrated that in a whole range of cardiovascular diseases such as acute or chronic heart failure [15,16], coronary artery disease or myocardial infarction [17,18], plasma concentrations of sST2 were markedly elevated compared to the normal population. Patients with severe AS also showed significantly elevated sST2 levels with prognostic significance [19].

Aim of the present study was to investigate how sST2 levels in patients with severe AS behaves in relation to the presence or absence of PH, in order to be able to make noninvasive statements about PH in dependence of sST2 plasma concentrations.

## 2. Methods

### 2.1. Patient Population

The overall cohort comprised 260 patients with severe AS, who could be included in the present study because a serum analytical determination of sST2 was performed pre-interventionally. The corresponding examinations were performed at the University Hospital Jena from 2010 to 2015. The study protocol was approved by the local ethics committee of the Friedrich Schiller University Jena (No.: 3237-09/11). Written informed consent for study participation was available from all patients. The study was conducted in accordance with principles of the Declaration of Helsinki and Good Clinical Practice. The indication for TAVR procedure was made in a multidisciplinary heart-team consisting of cardiologists and cardio surgeons. Follow-up was performed at 12 months after TAVR by outpatient examination.

### 2.2. Transthoracic Echocardiography

Transthoracic echocardiography was performed by experienced examiners using common ultrasound devices (iE33 and Epiq 5; Philips Healthcare, Hamburg, Germany). Severe AS was classified according to current valid guidelines of European Society for Cardiology measuring. Simpson’s method was applied to receive left ventricular ejection fraction (LVEF). To graduate mitral, aortic, and tricuspid valve regurgitation in minimal, mild (I), moderate (II) and severe (III) spectral and color-Doppler images were used. Maximal tricuspid regurgitant jet velocity combined with central venous pressure (diameter of inferior vena cava as important determinant) was used to calculate systolic pulmonary artery pressure (sPAP).

### 2.3. Right Heart Catheterization (RHC) Procedure

Patients with an echocardiographically sPAP ≥ 30 mmHg received a RHC procedure by using a standardized procedure via femoral vein access 2 to 4 weeks before TAVR procedure. Pressure curves were measured using fluid-filled catheters connected to pressure transducers. Right atrial pressure (mmHg), right ventricular pressure (mmHg), systolic artery pressure (mmHg), diastolic artery pressure (mmHg) and mean pulmonary artery pressure (mmHg) were recorded. Additionally, determinations of pulmonary capillary wedge pressure (mmHg) were performed. Cardiac output was assessed by using the modified Fick method with estimated oxygen consumption and was indexed to body surface area to calculate cardiac index. Metek Software (Metek, Elmshorn, Germany) was used for all calculations.

### 2.4. Hemodynamic Criterions of PH

PH was classified according to currently valid European Society for Cardiology guideline of 2015. PH was defined by a mPAP ≥ 25 mmHg, whereas a mPAP < 25 mmHg led to exclusion of PH. Patients with PH were further divided into pre-capillary PH (prec-PH) by PAWP ≤ 15 mmHg and post-capillary PH (postc-PH) by PAWP > 15 mmHg.

### 2.5. Analysis of sST2 by Enzyme-Linked Immunosorbent Assay

Blood samples were collected by puncture of a cubital vein using a vacuum-containing system one day before TAVR procedure. Collection tubes were centrifuged, and the obtained plasma samples were frozen at −80 °C until further measurements have been carried out. An enzyme-linked immunosorbent assay kit (Duoset DY206; R&D Systems, Minneapolis, MN, USA) was performed to determine plasma levels of sST2. Plasma samples and standard protein coated with respective capture antibody was added to the multi-well plate and incubated for two hours. After this incubation period plates were washed three times with buffer (Tween 20, Sigma Aldrich, St. Louis, MS, USA) and phosphate-buffered saline solutions. Then, a biotin-labelled secondary antibody was added, and the plates were incubated for another two hours. Plates were washed again, and Streptavidin-horseradish-peroxidase was added. Color reaction was obtained using tetramethylbenzidine (TMB; Sigma Aldrich, St. Louis, MS, USA). Optical density was measured at 450 nanometers on an ELISA plate-reader (iMark Microplate Absorbance Reader, Bio-Rad Laboratories, Vienna, Austria).

### 2.6. TAVR Procedure

TAVR procedure was performed as previously described [20]. Prosthesis of Edwards Lifesciences (Edwards SAPIEN, Irvine, CA, USA), of Medtronic (CoreValve, Dublin, Ireland) and of JenaValve Technology (JenaValve, Jena, Germany) were used for transfemoral approach. By transapical approach, JenaValve and Edwards SAPIEN were implanted.

### 2.7. Statistical Analysis

Statistical analysis was performed using SPSS 25 (SPSS Inc., Chicago, IL, USA).

First of all, the Kolmogorov–Smirnov test was applied to test variables for normal distribution. Normally distributed variables with nominal and ordinal scale level were specified as frequencies/percentages and normally distributed, metric variables by means of mean ± standard deviation. Non-normally distributed variables—especially sST2 as a biomarker—were presented as median with interquartile range.

For comparison of normally distributed data of two groups (No PH—PH), the chi-squared test was used for categorical variables and Student’s *t*-test for metric variables. ANOVA for metric data was applied, when more than two groups (No PH—prec-PH—postc-PH) were compared.

For non-normally distributed data of two groups Mann–Whitney-U test was performed, and for comparison of more than two groups, Kruskal–Wallis test with Dunn’s post hoc test was used.

To exclude possible influencing factors regarding the association between the presence of PH and sST2 level, first, a univariate, binary logistic regression analysis was completed. For better comparability, a z-transformation was absolved for metric data. Subsequently, multivariate, binary logistic regression was performed to assess independent factors regarding the prediction of PH. Therefore, covariates associated with detection of PH in the univariate analysis (*p <* 0.100) were entered and a backward variable elimination was carried out.

Correlation analyses were performed using Spearman’s rank-correlation coefficient to determine the strength between sST2 to further variables (age, diabetes, hypertension, etc.).

Univariate Cox proportional hazard regression model was used to calculate hazard ratio (HR) and 95% confidence interval (CI) for several influencing factors associated with 1-year-mortality in patients undergoing TAVR procedure. Again, the z-transform was applied for metric data. Afterwards, multivariate Cox regression was performed to assess independent predictors of mortality. Therefore, again, covariates associated with mortality in the univariate analysis (*p <* 0.100) were entered and a backward variable elimination was performed.

To determine an optimal cut-off value according to 1-year survival and sST2 serum level or mPAP ≥ 25 mmHg and sST2, area under the receiver operator characteristics (AUROC)-curves with area under the curve (AUC) and separate analysis of Youden index (YI) was performed.

Finally, Kaplan–Meier curves were carried out to detect overall survival of patients according to cut-off sST2 levels, which were obtained by ROC curves mentioned above.

A *p*-value of ≤0.050 was considered statistically significant.

## 3. Result

### 3.1. Study Cohort

A total of 260 patients with severe AS planned for TAVR procedure and sST2 biomarker determination pre-TAVR were enrolled in in the study. A total of 108 patients had no right heart catheter measurements, thus no statement could be made regarding PH according to current ESC guidelines. Of the remaining 152 study participants, 43 had mPAP < 25 mmHg, which led to exclusion of PH. In total, 109 patients showed PH with an mPAP ≥ 25 mmHg, of which 11 showed a PAWP of ≤15 mmHg and 98 showed a PAWP of >15 mmHg, formally fulfilling the criteria of prec-PH and postc-PH, respectively (Figure 1).

### 3.2. Baseline Characteristic of Study Participants

Baseline characteristics were divided into “PH not deducible” (no mPAP data), “No PH” (mPAP < 25 mmHg) and “PH” (mPAP ≥ 25 mmHg) (Table 1).

Of course, the major differences between non-PH and PH cohorts were achieved in the measured right heart catheterization data, where the significance level was <0.001. Additionally, highly significant differences between non-PH and PH cohorts were observed with respect to echocardiographically determined sPAP (33.43 ± 7.77 mmHg vs. 45.23 ± 14.55 mmHg) and CRP (3.10 ± 7.50 mg/L vs. 7.30 ± 18.10 mg/L).

### 3.3. Biomarker Concentrations

In the non-PH group with severe AS (*n* = 43), sST2 was 5233.95 ± 3621.56 pg/mL and in the PH group (*n* = 109) it was 8239.14 ± 8187 pg/mL, which resulted in a significant difference of *p* = 0.006 (Table 1).

Regarding a further subdivision of PH, patients with prec-PH showed an sST2 plasma level of 5274.85 ± 6864.82 pg/mL and in patients with postc-PH of 8239.14 ± 9618.89 pg/mL. A relevant significance level of *p* = 0.003 was observed between the non-PH and postc-PH groups, whereas a *p* = 0.918 was detected between non-PH and prec-PH, and a *p* = 0.109 was found between prec-PH and postc-PH (Figure 2).

### 3.4. Binary, Logistic Regression Analysis

In order to verify a relevant statistical relationship between the presence of PH and other factors (especially sST2), a univariate as well as a multivariate binary logistic regression analysis was performed (Table 2).

In the univariate analysis, AV Vmax, AV dpmean, AVdpmax, mitral insufficiency ≥ II°, tricuspid insufficiency ≥ II° and sST2 showed a relevant association (*p* < 0.100), so multivariate analysis was performed with these variables. Only sST2 was found to have a significant *p*-value of 0.034.

### 3.5. Correlation Analysis

To investigate relationships between sST2 plasma level and other patients’ characteristics, Spearman correlation analysis was performed (Table 3).

It was striking that both the overall cohort and the PH group showed positive correlations with regard to the collected right heart catheter data. For better visualization, the correlation between sST2 and mPAP and sST2 and PAWP was represented as a scatterplot depending on the presence or absence of a PH (Figure 3). NYHA score ≥ III° (PH group—rs: 0.210, *p* = 0.030), LVEDD (PH group—rs: 0.403, *p* < 0.001) and LVESD (PH group—rs: 0.302, *p* = 0.006) diameters, LVEDP (PH group—rs: 0.286, *p* = 0.007), and CRP (PH group—rs: 0.299, *p* = 0.003) also correlated positively with serum sST2 level. Inverse correlations could be observed in LVEF (PH group—rs: −0.374, *p* < 0.001) and EuroScore (overall cohort—rs: −0.045, *p* = 0.044; PH group—rs: −0.256, *p* = 0.010). Echocardiographically assessed TAPSE as a measure of right ventricular function showed only an inverse trend (PH group—rs: −0.194, *p* = 0.058).

In the non-PH group, positive correlations were seen with respect to LVEDD diameter (non-PH group—rs: 0.323, *p* = 0.034) and echocardiographically assessed mitral (non-PH group—rs: 0.330, *p* = 0.033) and tricuspid insufficiencies ≥ II° (non-PH group—rs: 0.455, *p* = 0.003).

### 3.6. Cox Proportional Hazard Regression

To investigate several influencing variables concerning 1-year mortality after TAVR, a univariate and multivariate Cox proportional hazard regression was presented (Table 4).

The result of univariate analyses showed agreement (*p* < 0.100) with right heart catheterization data (RA, RV, sPAP, mPAP, dPAP, PAWP), with echocardiographic data (LVEF, LVEDD, sPAP, mitral and tricuspid insufficiency ≥ II°), with laboratory chemistry data (CRP, sST2), with concomitant diseases (BMI, myocardial infarction, diabetes mellitus), and with STS score. After inclusion of these data in a multivariate analysis, STS score, diabetes mellitus, LVEDD, mPAP, dPAP, PCWP, and sST2 (*p* = 0.006) remained as independent factors.

### 3.7. AUROC Results

To analyze sST2 as a potential biomarker for prediction of mortality in PH patients with severe AS before TAVR, AUROC-curves regarding sST2 plasma level concentration in dependency of 1-year mortality and in dependence of mPAP ≥ 25 mmHg were figured out. Therefore, AUC, cut-off values with YI, as well as sensitivity and specificity were extracted in addition to ROC curves (Figure 4).

This analysis identified a sST2 plasma level of 5521.15 pg/mL as an optimal cut-off value concerning 1-year mortality for the non-PH group (AUC 0.794; 95%CI 0.634–0.954; *p* = 0.015; YI 0.52; sensitivity 0.86; specificity 0.67) (Figure 3A), whereas the cut off value of the PH group was at a considerably higher sST2 plasma level of 10,268.78 pg/mL (AUC 0.628; 95%CI 0.503–0.754; *p* = 0.038; YI 0.29; sensitivity 0.55; specificity 0.74) (Figure 4B).

In addition, it was analyzed where the corresponding sST2 cut-off value is, from which a possible PH can be inferred. For this purpose, an mPAP ≥ 25 mmHg was used according to ESC guidelines. An optimal cut-off value was at 6990.12 pg/mL (AUC 0.643; 95%CI 0.549–0.737; *p* = 0.006; YI 0.31; sensitivity 0.57; specificity 0.74) (Figure 4C).

### 3.8. Scatterplot

A graphical representation of the sST2 value pairs as a function of mPAP is shown in Figure 5. In this regard, a cutoff value for an mPAP of 25 mmHg was defined for the presence (≥25 mmHg) or absence (<25 mmHg) of PH. The aforementioned determined sST2 cut-off value of 6990.12 pg/mL was used to divide a total of four groups: group I (blue dots; *n* = 32) showed an mPAP < 25 mmHg and sST2 < 6990.12 pg/mL, group II (red dots; *n* = 47) an mPAP ≥ 25 mmHg and sST2 < 6990.12 pg/mL, group III (green dots; *n* = 62) an mPAP ≥ 25 mmHg and sST2 ≥ 6990.12 pg/mL, and group IV (purple dots; *n* = 11) an mPAP < 25 mmHg and sST2 ≥ 6990.12 pg/mL.

### 3.9. Kaplan–Meier Curves

Kaplan-Maier curves were performed with regard to 1-year survival as a function of plasma level concentration of sST2 (Figure 6).

For Figure 6A, the previously established four groups (Figure 5) were compared with each other, and the significance was analyzed with a log-rank test. With the exception of comparison between group I and group III, there were no relevant differences in 1-year survival between the different groups. Accordingly, patients without evidence of PH and an sST2 concentration < 6990.12 pg/mL (group I) showed significantly lower 1-year mortality in contrast to patients with evidence of PH and sST2 ≥ 6990 pg/mL (group III) (log-rank test *p* = 0.027). A total of 12.5% of study participants from group I died within 1 year, whereas 34.4% from group III were no longer alive after 1 year. Despite the lack of significance, the Kaplan–Meier curves obtained showed a general tendency for mortality rates to increase in the presence of PH or an sST2 level ≥ 6990.12 pg/mL or a combination of both.

For Figure 6B, the sST2 cut-off value was raised to 10,268.78 pg/mL, as this corresponded to the 1-year mortality value of PH patients (see AUROC results). The general log-rank test was already significant here with *p* = 0.005. Specified log-rank analyses showed further significant values between group I (no PH + sST2 < 10,268.78 pg/mL) and group III (PH + sST2 ≥ 10,268.78 pg/mL) with *p* = 0.001 and between group II (PH + sST2 < 10268.78 pg/mL) and group III with *p* = 0.009.

### 3.10. Sensitivity and Specificity of sST2 in Prediction of PH

It was demonstrated that an sST2 level of 6990.12 pg/mL had a significantly increased probability of finding a patient with RHC-verified PH. Overall, 62/152 study participants exhibited this constellation, with 34.4% dying within 1 year. The highest mortality rate manifested in a combination of high mPAP and high plasma sST2 concentrations. In total, 32/152 patients with the exact opposite constellation—no PH and sST2 < 6990.12 pg/mL—had significantly better survival.

However, 47/152 patients, representing a percentage of 30.9%, also showed mPAP ≥ 25 mmHg and sST2 < 6990.12 pg/mL. From this group, 21.7% died within a 1-year follow-up, although a direct comparison regarding mortality of patients with high mPAP and high sST2 did not yield significant differences. In addition, 11/152 patients and thus 7.2% could be detected with sST2 data ≥ 6990.12 pg/mL and normative mPAP data; again, 27.3% of these were no longer alive after one year.

For serum sST2 concentrations ≥ 6990 pg/mL, sensitivity of 84.9% (62/62 + 11) to actually correctly identify a patient with PH in the RHC data is relatively high. However, at sST2 serum concentrations of <6990 pg/mL, the specificity of 40.5% is very low (32/32 + 47) to select a patient without PH. The false positive rate was 15.1% (11/11 + 62) and the false positive rate was 59.5% (47/47 + 32). In summary, in the present study, there would be an error rate of 38.8% (11 + 47/152) if an sST2 cut-off value of nearly 7000 pg/mL were used in isolation to make a definitive statement about the presence or absence of PH.

## 4. Discussion

The fact that sST2 is a pioneering cardiac biomarker for predicting the extent and severity of many cardiac and cardiovascular diseases is no longer news. As early as 2013, sST2 was recommended by the American College of Cardiology and the American Heart Association as a predictor of hospitalization and death in patients with acute and chronic heart failure [21]. Other larger, clinical studies have since been completed, which also demonstrated elevated sST2 levels in patients with congenital [22] and ischemic [23] heart disease as well as with acute coronary syndrome [24,25]. Additionally, in valvular cardiomyopathies, especially in severe AS, sST2 could be positioned as a relevant biomarker for predictions of mortality and as a relevant criterion for adequate risk stratification [26]. Studies of sST2 and severe AS have never investigated the association between serum sST2 concentration and the most important and survival-limiting comorbidity—pulmonary hypertension. The data presented here should help, at least in part, to fill this scientific gap.

### 4.1. Double “Stress” for the Heart in AS with PH?

We demonstrated that pre-TAVR sST2 concentrations showed significant differences between severe AS patients with and without PH. Already in numerous studies [27], significantly elevated plasma sST2 concentrations were found in patients with severe AS, compared with the normal collective. Here, the pressure load with consecutive myocardial hypertrophy and corresponding remodeling processes can be regarded as causative. The association of elevated sST2 and pulmonary hypertension has also been reported recently [28,29]. Pathognomonic for this is probably the pulmonary vascular remodeling, which in turn leads to a right heart strain and an associated strain-mediated myocardial release. In our study, using TAPSE as an important measure of right ventricular function in patients with PH, we therefore found a trend (*p* = 0.058) regarding an inverse correlation with sST2 level. Finally, with the current state of research, it cannot be definitely excluded that sST2 is not additionally released by endothelial cells of the stressed pulmonary vasculature, which are exposed to considerable inflammatory processes [30,31].

However, current studies mainly concentrate on either severe AS or pulmonary hypertension. According to present studies, PH is found in 10 to 42% of patients with echocardiographically verified severe AS [32]. Therefore, it is not surprising that AS patients with RHC-confirmed PH have significantly higher sST2 levels than AS patients with normotensive RHC data and solitary left ventricular strain, because of the dual stress of elevated pressures in both the left and the right cardiovascular system. We were ultimately able to strengthen this hypothesis by the multivariate binary logistic regression performed, leaving sST2 as the only independent factor with *p* = 0.034.

### 4.2. Can sST2 as a Biomarker Predict or Even Replace RHC Measurements?

The correlation analyses performed in the present study showed numerous similarities to previously published data. In the overall cohort as well as in the PH group an inverse correlation for the investigation between sST2-level and lower LVEF values—synonymous for heart failure with reduced ejection fraction—was manifested, which was similarly described in the corresponding study of Lancelotti et al. [33]. In addition, Binas et al. [34] demonstrated increased sST2 levels in patients with dilated cardiomyopathy, with concordance regarding the LVEDD and LVESD correlation analyses described here. According to current data, progressive dyspneic symptoms [35] also lead to an increase in sST2 levels, which is why it should not be surprising that positive correlations were also found in here with higher NYHA classification in both the overall cohort and the PH group. There are also data from Galeone et al. [36] showing increased mortality in patients with valvular cardiomyopathy and corresponding sST2 values. There was a positive correlation in the presence of mitral valve and tricuspid valve regurgitation ≥ II° in the overall cohort and the non-PH group. Worth mentioning, however, is that echocardiographically obtained parameters for AS (AV Vmax, Pmean, Pmax, AVA) did not correlate with the sST2 level in our study, so that it can only be speculated here that the extent of myocardial hypertrophy or fibrosis and thus secretion of sST2 is not directly related to criteria of pressure load, at least not in patients with severe AS.

Particular attention should be paid to the association between sST2 and available right heart catheterization data. Positive correlations in this regard have already been described by Banaszkiewicz et al. [37,38], who examined various subgroupings of PH for this association specifically. In our study, post-capillary PH patients showed a significant difference in terms of serum sST2 concentration in contrast to the patients without PH. In turn, patients with RHC determined pre-capillary PH demonstrated no significant difference in terms of serum sST2 concentration. Considering the correlation analyses (Table 3) with additional graphical representation in the form of a scatterplot (Figure 3), it can be hypothesized that sST2 is mainly associated with high mPAP as well as high PAWP data. However, it should be noted that the number of patients with pre-capillary PH was very low (*n* = 11), because patients with severe AS generally develop post-capillary PH in a high percentage. Accordingly, no hasty conclusions should be made about the relationship between pre-capillary PH and sST2 concentrations in the present collective. Nevertheless, it cannot be denied that consequent pressure and volume overload of the left ventricle in the setting of severe AS may, on the one hand, increase cardiomyocyte secretion of sST2 and, on the other hand, increase right heart catheter data such as mPAP and PAWP due to the same pathophysiology. Fibrotic disease of the lung associated with pre-capillary PH may lead to markedly reduced sST2 release due to lack of myocardial stretch.

### 4.3. High sST2 in AS Equivalent to PH and Increased Mortality?

It is now well known that PH as an accompanying comorbidity to AS is associated with increased mortality after TAVR [39]. However, in the current era, RHC is no longer routinely performed in TAVR evaluation, but rather the absence or presence of PH is estimated echocardiographically with sPAP. Therefore, the possibility of a noninvasive statement about a potential presence of pulmonary hypertension by means of biomarkers becomes more and more important in this context.

In the performed Cox regression analysis—especially in the multivariate analysis—it was shown that sST2 was an independent marker for the prediction of 1-year mortality in addition to the invasive, collected right heart catheter data (*p* = 0.006). In contrast, other non-invasive, echocardiographic data such as sPAP (*p* = 0.419), mitral insufficiency ≥ II° (*p* = 0.279) and tricuspid insufficiency ≥ II° (*p* = 0.483) were excluded. These results could be supported by Gül et al. [40] in their study of heart failure patients, who showed equally significant values regarding sST2 and non-significant values regarding echocardiographic data in terms of their Cox regression.

Despite the satisfactory data, it should not be disregarded that the determined sST2 cut off value of just under 7000 pg/mL for noninvasive detection of PH has an error rate of approximately 40%. This again makes clear that a singular biomarker determination, e.g., by means of sST2 is not sufficient to detect PH. The aim of further studies must therefore be to develop a risk score for PH similar to the EuroScore or the STS score based on clinical factors that are easy to collect (echocardiographic data, other laboratory chemistry data besides sST2, clinical data such as 6 min walk test, lung function, etc.).

## 5. Conclusions

sST2 remains a relevant biomarker in patients with severe AS. In patients with high sST2 levels, a high percentage of almost 85% can be assumed to have relevant PH in addition to AS. In contrast, however, there is only a 40.5% specificity. A risk score regarding the presence of survival-limiting PH should be enforced in further studies to better assess which patients can definitely benefit from TAVR.

## 6. Limitation

The present studied was based on data of a single-center with a small cohort. sST2 levels were only detected at baseline without a statement regarding sST2 expression after TAVR procedure. Additionally, technical pitfalls in RHC measurements should always be conceded in the context of hemodynamic measurements, even if examinations were performed by experienced clinical investigators.

## Figures and Tables

**Figure 1 life-12-00389-f001:**
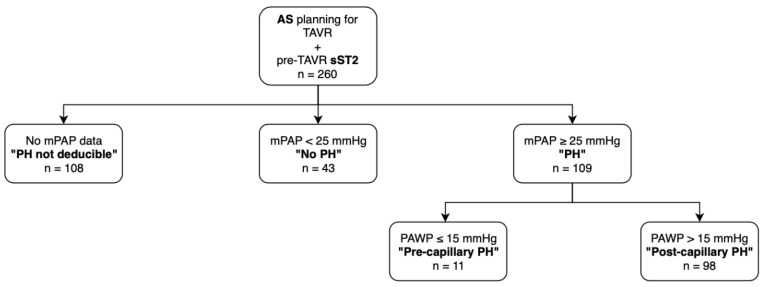
Patient disposition in study cohort. AS: aortic stenosis; TAVR: transcatheter aortic valve replacement; sST2: soluble suppression of tumorigenicity 2; mPAP: mean pulmonary arterial pressure; PH: pulmonary hypertension; PAWP: pulmonary capillary wedge pressure.

**Figure 2 life-12-00389-f002:**
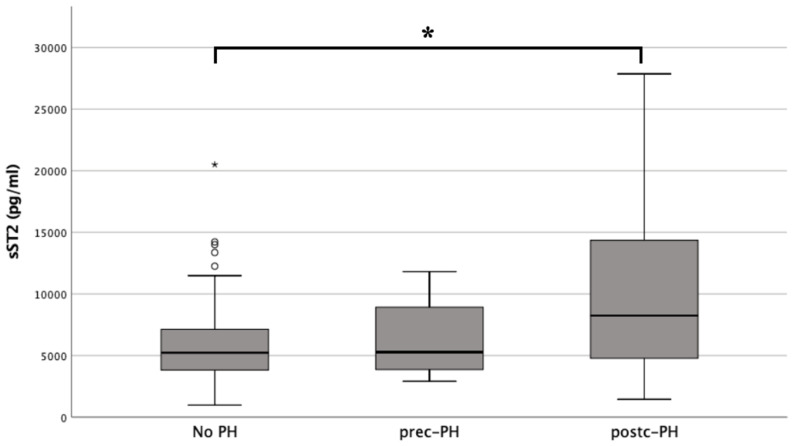
Results of sST2 analysis in dependence of PH presence or PH absence No PH vs. prec-PH: *p* = 0.918 No PH vs. post-PH: *p* = 0.003 (* *p* ≤ 0.050) Prec-PH vs. postc-PH: *p* = 0.109. sST2: soluble suppression of tumorigenicity 2; PH: pulmonary hypertension; prec-PH: precapillary PH, postc-PH: postcapillary PH.

**Figure 3 life-12-00389-f003:**
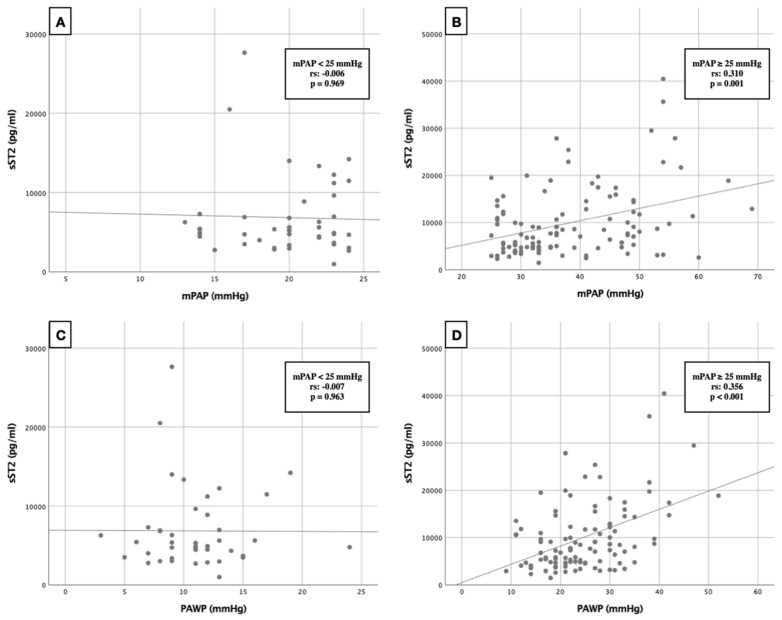
Scatterplot of correlation analysis between sST2 and RHC data. (**A**,**B**) Correlation between sST2 and mPAP in dependence of presence or absence of PH. (**C**,**D**) Correlation between sST2 and PAWP in dependence of presence or absence of PH. sST2: soluble suppression of tumorigenicity 2; mPAP: mean pulmonary arterial pressure; PAWP: pulmonary capillary wedge pressure; rs: correlation coefficient of Spearman.

**Figure 4 life-12-00389-f004:**
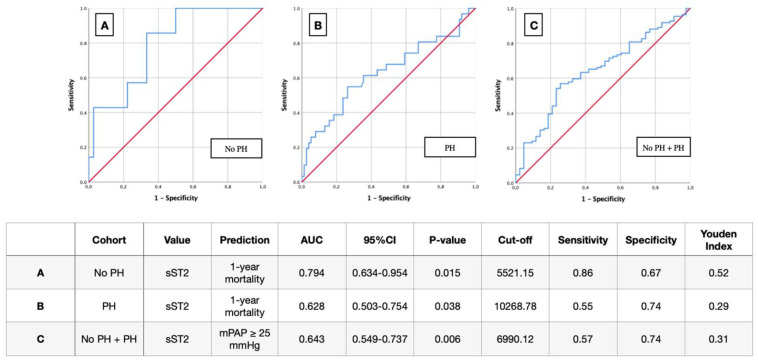
AUROC curves, cut-off values and Youden Index for prediction of 1-year mortality (**A**,**B**) and for prediction of presence or absence of PH (**C**) according to sST2 serum levels. PH: pulmonary hypertension; sST2: soluble suppression of tumorigenicity 2; mPAP: mean pulmonary arterial pressure; AUC: area under the curve; CI: confidence interval.

**Figure 5 life-12-00389-f005:**
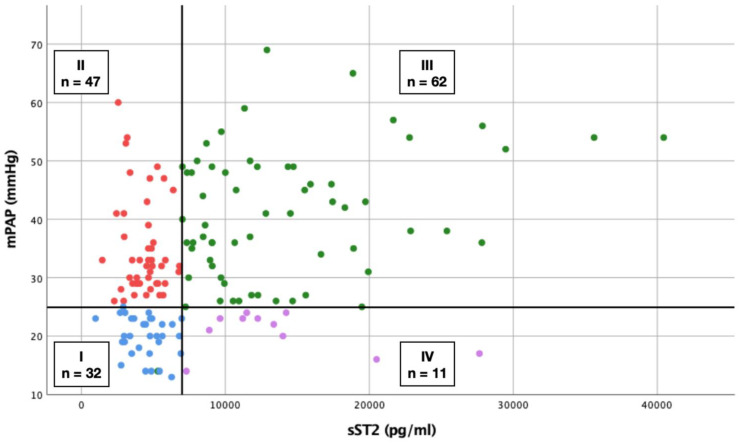
Scatterplot of sST2 value pairs as a function of mPAP Group I (blue dots): mPAP < 25 mmHg + sST2 < 6990.12 pg/mL Group II (red dots): mPAP ≥ 25 mmHg + sST2 < 6990.12 pg/mL Group III (green dots): mPAP ≥ 25 mmHg + sST2 ≥ 6990.12 pg/mL Group IV (purple dots): mPAP < 25 mmHg + sST2 ≥ 6990.12 pg/mL. sST2: soluble suppression of tumorigenicity 2; mPAP: mean pulmonary arterial pressure.

**Figure 6 life-12-00389-f006:**
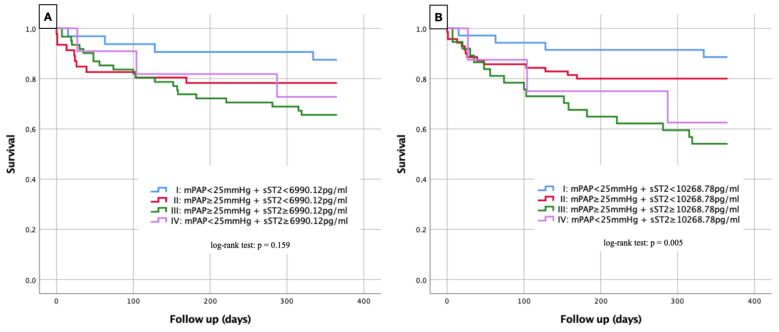
Kaplan–Meier curves for detection of 1-year survival in dependence of several risk groups. (**A**) Group I (blue line): mPAP < 25 mmHg + sST2 < 6990.12 pg/mL; Group II (red line): mPAP ≥ 25 mmHg + sST2 < 6990.12 pg/mL; Group III (green line): mPAP ≥ 25 mmHg + sST2 ≥ 6990.12 pg/mL; Group IV (purple line): mPAP < 25 mmHg + sST2 ≥ 6990.12 pg/mL. Log-rank test Group I vs. Group II: *p* = 0.266; Group I vs. Group III: *p* = 0.027; Group I vs. Group IV: *p* = 0.249; Group II vs. Group III: *p* = 0.233; Group II vs. Group IV: *p* = 0.818; Group III vs. Group IV: *p* = 0.633. (**B**) Group I (blue line): mPAP < 25 mmHg + sST2 < 10,268.78 pg/mL; Group II (red line): mPAP ≥ 25 mmHg + sST2 < 10,268.78 pg/mL; Group III (green line): mPAP ≥ 25 mmHg + sST2 ≥ 10,268.78 pg/mL; Group IV (purple line): mPAP < 25 mmHg + sST2 ≥ 10,268.78 pg/mL. Log-rank test Group I vs. Group II: *p* = 0.255; Group I vs. Group III: *p* = 0.001; Group I vs. Group IV: *p* = 0.063; Group II vs. Group III: *p* = 0.009; Group II vs. Group IV: *p* = 0.326; Group III vs. Group IV: *p* = 0.651. sST2: soluble suppression of tumorigenicity 2; mPAP: mean pulmonary arterial pressure.

**Table 1 life-12-00389-t001:** Clinical data, concomitant diseases, echocardiographic measurements, invasive hemodynamic profile, laboratory data and procedural data of study cohort. * *p*-values comparing three groups (“PH not deducible”, “No PH” and “PH”) were calculated. ** *p*-values comparing two groups (“No PH” and “PH”) were calculated. PH: pulmonary hypertension; BMI: body mass index; CVD: cardiovascular disease; COPD: chronic obstructive pulmonary disease; LVEF: left ventricular ejection fraction; LVEDD: left ventricular end diastolic diameter; LVESD: left ventricular end systolic diameter; sPAP: systolic pulmonary arterial pressure; AVA: aortic valve area; AV max: maximal velocity over aortic valve; AV dpmean: mean pressure gradient over aortic valve; AV dpmax: maximal pressure gradient over aortic valve; RA: right atrium pressure; RV: right ventricular pressure; mPAP: mean pulmonary arterial pressure; dPAP: diastolic pulmonary arterial pressure; PAWP: pulmonary arterial wedge pressure; LVEDP: left ventricular end-diastolic pressure; CRP: c-reactive protein; Hb: hemoglobin; sST2: soluble suppression of tumorigenicity 2; BNP: brain natriuretic peptide.

	PH Not Deducible (No mPAP Data) *n* = 108		No PH (mPAP < 25 mmHg) *n* = 43		PH (mPAP ≥ 25 mmHg) *n* = 109		All Groups	No PH vs. PH
Clinical data	mean	SD	mean	SD	mean	SD	*p*-value *	*p*-value **
Age (years)	80.49	6.46	80.91	6.87	81.05	7.6	0.924	0.917
Weight (kg)	75.06	15.25	72.79	13.45	76	13.35	0.056	0.185
Height (cm)	164.77	9.28	163.14	9.82	165.52	8.5	0.119	0.139
BMI (kg/m^2^)	27.63	5.2	27.32	4.21	27.74	4.47	0.356	0.597
NYHA	3.11	0.61	2.88	0.64	3.15	0.56	**0.028**	**0.019**
STSScore	2.1	2.22	3.73	2.44	4.08	2.49	0.08	0.439
EuroScore	27.12	17.04	21.05	13.54	26.14	14.3	0.686	0.055
Concomitant Disease	%		%		%			
Diabetes mellitus	63.3		55.8		60.6		0.374	0.593
Arterial Hypertension	91.1		90.7		94.5		0.896	0.395
CVD-1 vessel	18.9		18.6		27.5		0.665	0.253
CVD-2 vessels	9.2		9.3		10.1		0.251	0.883
CVD-3 vessels	18.5		14		17.4		0.355	0.602
COPD	23.1		18.6		26.6		0.181	0.301
Myocardial infarction	13		14		14.8		0.273	0.892
Stroke	15.1		9.3		15.6		0.801	0.311
Echocardiography	mean	SD	mean	SD	mean	SD		
LVEF (%)	54.02	17.55	60.71	16.68	56.17	18.08	**0.025**	0.161
LVEDD (mm)	49.04	8.31	47.79	8.2	49.18	7.59	0.082	0.328
LVESD (mm)	33.27	9.54	30.06	10.21	32.26	9.29	0.054	0.253
sPAP (mmHg)	29.52	6.45	33.43	7.77	45.23	14.55	**<0.001**	**<0.001**
AVA (cm^2^)	0.66	0.2	0.66	0.22	0.65	0.17	0.424	0.784
AV Vmax (m/s)	4.24	0.7	5.41	6.14	4.16	0.68	**0.01**	0.189
AV dPmean (mmHg)	43.85	15.9	50.92	16.73	44.85	16.66	**0.012**	**0.048**
AV dPmax (mmHg)	73.09	23.87	82.39	23.1	74.03	25.74	**0.022**	0.067
TAPSE (mm)	22.2	5.7	21.3	5.2	17.7	4.4	0.098	0.072
RHC & LHC data	mean	SD	mean	SD	mean	SD		
RA (mmHg)			5.95	2.79	12.72	6.59		**<0.001**
RV (mmHg)			4.93	3.84	11.42	7.15		**<0.001**
sPAP (mmHg)			34.02	6.87	59.95	16.6		**<0.001**
mPAP (mmHg)			19.88	3.43	38.35	10.27		**<0.001**
dPAP (mmHg)			9.91	3.5	22.61	7.7		**<0.001**
PAWP (mmHg)			11	3.77	24.68	8.24		**<0.001**
LVEDP (mmHg)	21.54	7.04	19.79	6.06	21.77	7.74	0.218	0.163
Laboratory data	median	IQR	median	IQR	median	IQR		
Creatinine (µmol/L)	111.5	64.75	87	36	102	65.25	0.051	0.309
CRP (mg/L)	7.35	20.5	3.1	7.5	7.3	18.1	**<0.001**	**0.001**
Hb (mmol/L)	7.75	1.53	7.7	1.4	7.55	1.47	0.559	0.205
sST2 (pg/mL)	5639.73	5521.99	5233.95	3631.56	8239.14	8187	**0.002**	**0.006**
BNP (pg/mL)	444.5	892.75	660	2029.5	854	2682.75	0.151	0.577
Procedural data	%		%		%			
Transfemoral	72.6		74.4		77.1		0.787	0.73
Edwards	70.3		65.1		63.3		**0.016**	0.834
CoreValve	14.3		11.6		20		0.056	0.157
JenaValve	10		23.3		16.7		0.195	0.347
Vascular Complications	8.9		9.3		10.1		0.71	0.883

**Table 2 life-12-00389-t002:** Univariate and multivariate, binary, logistic regression analysis detecting predictors of PH via sPAP ≥ 25 mmHg. BMI: body mass index; CVD: cardiovascular disease; COPD: chronic obstructive pulmonary disease; LVEF: left ventricular ejection fraction; LVEDD: left ventricular end diastolic diameter; LVESD: left ventricular end systolic diameter; AVA: aortic valve area; AV max: maximal velocity over aortic valve; AV dpmean: mean pressure gradient over aortic valve; AV dpmax: maximal pressure gradient over aortic valve; CRP: c-reactive protein; Hb: hemoglobin; sST2: soluble suppression of tumorigenicity 2.

mPAP ≥ 25 mmHg Binary Logistic Regression	Univariate		Multivariate	
	Hazard Ratio (95% CI)	*p*-Value	Hazard Ratio (95% CI)	*p*-Value
Age	1.018 (0.735–1.410)	0.916		
Weight	1.301 (0.882–1.919)	0.185		
BMI	1.113 (0.750–1.654)	0.595		
Diabetes mellitus	0.823 (0.403–1.681)	0.593		
Arterial Hypertension	0.568 (0.152–2.121)	0.400		
Cardiovascular Disease (all)	0.701 (0.334–1.475)	0.349		
CVD-2 vessels	0.914 (0.274–3.043)	0.883		
CVD-3 vessels	0.768 (0.284–2.076)	0.603		
COPD	0.631 (0.262–1.517)	0.303		
Myocardial infarction	0.932 (0.339–2.567)	0.892		
Stroke	0.555 (0.175–1.756)	0.316		
LVEF	0.762 (0.520–1.115)	0.162		
LVEDD	1.201 (0.832–1.734)	0.327		
LVESD	1.274 (0.841–1.928)	0.253		
AVA	0.942 (0.647–1.372)	0.756		
AV Vmax	0.136 (0.029–0.650)	0.012	0.258 (0.044–1.497)	0.131
AV dpmean	0.707 (0.500–1.002)	0.051	1.452 (0.414–5.086)	0.560
AV dpmax	0.726 (0.511–1.031)	0.073	0.744 (0.447–1.238)	0.255
Mitral insufficiency ≥ II°	0.468 (0.223–0.979)	0.044	1.161 (0.493–2.735)	0.733
Tricuspid insufficiency ≥ II°	0.471 (0.223–0.996)	0.049	0.540 (0.241–1.207)	0.133
Creatinine	1.214 (0.786–1.875)	0.383		
Hb	0.801 (0.555–1.155)	0.235		
sST2	1.748 (1.107–2.760)	0.017	1.697 (1.040–2.768)	0.034

**Table 3 life-12-00389-t003:** Tabular overview of correlation analysis with regard to various clinical characteristics and hemodynamic measurements. rs: correlation coefficient of Spearman; PH: pulmonary hypertension; BMI: body mass index; CVD: cardiovascular disease; COPD: chronic obstructive pulmonary disease; LVEF: left ventricular ejection fraction; LVEDD: left ventricular end diastolic diameter; LVESD: left ventricular end systolic diameter; sPAP: systolic pulmonary arterial pressure; AVA: aortic valve area; AV max: maximal velocity over aortic valve; AV dpmean: mean pressure gradient over aortic valve; AV dpmax: maximal pressure gradient over aortic valve; RA: right atrium pressure; RV: right ventricular pressure; mPAP: mean pulmonary arterial pressure; dPAP: diastolic pulmonary arterial pressure; PAWP: pulmonary arterial wedge pressure; LVEDP: left ventricular end-diastolic pressure; CRP: c-reactive protein; Hb: hemoglobin; BNP: brain natriuretic peptide.

	Overall Cohort mPAP + No mPAP Data		sST2—No PH mPAP < 25 mmHg		sST2—PH mPAP ≥ 25 mmHg	
	rs	*p*	rs	*p*	rs	*p*
Age	0.006	0.923	0.152	0.329	−0.101	0.294
Weight	0.116	0.063	0.114	0.467	0.054	0.575
Height	0.229	**<0.001**	0.259	0.094	0.232	**0.015**
BMI	−0.044	0.476	−0.075	0.631	−0.07	0.471
Survival time (1 year)	−0.233	**<0.001**	−0.379	**0.012**	−0.165	0.089
NYHA ≥ III°	0.17	**0.007**	0.056	0.729	0.21	**0.03**
STSScore	−0.019	0.815	0.08	0.611	−0.078	0.421
EuroScore	−0.045	**0.044**	0.083	0.612	−0.256	**0.01**
Diabetes mellitus	0.019	0.757	0.279	0.07	−0.088	0.363
Arterial Hypertension	−0.076	0.225	−0.155	0.321	−0.093	0.334
Cardiovascular Disease (all)	0.008	0.899	0.256	0.098	−0.079	0.412
CVD-2 vessel	−0.008	0.897	0.09	0.565	−0.009	0.928
CVD-3 vessel	0.047	0.447	0.168	0.283	0.136	0.158
COPD	0.171	**0.006**	0.144	0.355	0.092	0.339
Myocardial infarction	0.101	0.104	0.297	0.053	0.165	0.088
Stroke	0.009	0.884	0.135	0.386	0.055	0.572
LVEF	−0.217	**0.001**	0.019	0.903	−0.374	**<0.001**
LVEDD	0.265	**<0.001**	0.323	**0.034**	0.403	**<0.001**
LVESD	0.251	**0.001**	0.266	0.117	0.302	**0.006**
sPAP (echocardiography)	0.274	**<0.001**	0.112	0.508	0.175	0.098
AVA	−0.071	0.281	0.222	0.162	−0.134	0.181
AV Vmax	−0.075	0.252	−0.152	0.329	−0.185	0.059
AV dPmean	−0.029	0.65	−0.151	0.332	−0.171	0.082
TAPSE	−0.102	0.108	−0.072	0.644	−0.194	0.058
Mitral insufficiency ≥ II°	0.175	**0.006**	0.33	**0.033**	0.063	0.521
Tricuspid insufficiency ≥ II°	0.211	**0.001**	0.455	**0.003**	0.074	0.469
RA	0.431	**<0.001**	0.401	**0.008**	0.38	**<0.001**
RV	0.276	**0.001**	0.222	0.158	0.221	**0.027**
sPAP (RHC)	0.312	**<0.001**	−0.008	0.959	0.282	**0.003**
mPAP	0.332	**<0.001**	−0.006	0.969	0.31	**0.001**
dPAP	0.342	**<0.001**	−0.182	0.244	0.347	**<0.001**
PAWP	0.358	**<0.001**	−0.007	0.963	0.356	**<0.001**
LVEDP	0.148	**0.031**	−0.028	0.867	0.286	**0.007**
Creatinine	0.235	**<0.001**	0.42	**0.007**	0.131	0.187
CRP	0.385	**<0.001**	0.272	0.094	0.299	**0.003**
Hb	−0.137	**0.038**	−0.23	0.158	0.058	0.574
BNP	0.033	0.79	0.147	0.587	0.133	0.68

**Table 4 life-12-00389-t004:** Univariate and multivariate Cox regression analysis detecting predictors of 1-year mortality. BMI: body mass index; CVD: cardiovascular disease; COPD: chronic obstructive pulmonary disease; LVEF: left ventricular ejection fraction; LVEDD: left ventricular end diastolic diameter; sPAP: systolic pulmonary arterial pressure; AVA: aortic valve area; AV max: maximal velocity over aortic valve; AV dpmean: mean pressure gradient over aortic valve; RA: right atrium pressure; RV: right ventricular pressure; mPAP: mean pulmonary arterial pressure; dPAP: diastolic pulmonary arterial pressure; PAWP: pulmonary arterial wedge pressure; LVEDP: left ventricular end-diastolic pressure; CRP: c-reactive protein; Hb: hemoglobin; BNP: brain natriuretic peptide; sST2: soluble suppression of tumorigenicity 2.

1-Year Mortality Cox Regression Analysis	Univariate		Multivariate	
	Hazard Ratio (95% CI)	*p*-Value	Hazard Ratio (95% CI)	*p*-Value
Age	1.003 (0.739–1.361)	0.985	-	
Weight	0.859 (0.609–1.211)	0.386		
BMI	0.688 (0.463–1.023)	0.065	0.790 (0.516–1.211)	0.280
NYHA ≥ III	0.657 (0.233–1.852)	0.427		
STS-Score	1.395 (1.123–1.733)	0.003	1.397 (1.073–1.821)	0.013
EuroScore	0.923 (0.652–1.306)	0.650		
Diabetes mellitus	0.547 (0.271–1.103)	0.092	0.304 (0.119–0.778)	0.013
Arterial Hypertension	0.345 (0.047–2.516)	0.294		
Cardiovascular Disease (all)	0.891 (0.442–1.796)	0.746		
CVD-2 vessels	1.345 (0.414–4.374)	0.622		
CVD-3 vessels	1.494 (0.685–3.259)	0.313		
COPD	0.651 (0.328–1.290)	0.218		
Myocardial infarction	0.476 (0.225–1.006)	0.052	0.904 (0.443–1.842)	0.780
LVEF	0.770 (0.570–1.039)	0.088	1.573 (0.891–2.777)	0.118
LVEDD	1.485 (1.163–1.896)	0.002	1.659 (1.158–2.377)	0.006
sPAP (echocardiography)	1.474 (1.105–1.967)	0.008	0.772 (0.412–1.446)	0.419
AVA	0.885 (0.618–1.268)	0.505		
AV Vmax	0.394 (0.105–1.483)	0.168		
AV dpmean	0.778 (0.550–1.101)	0.157		
Mitral insufficiency ≥ II°	0.514 (0.265–1.000)	0.050	1.627 (0.674–3.932)	0.279
Tricuspid insufficiency ≥ II°	0.408 (0.195–0.855)	0.017	0.677 (0.228–2.011)	0.483
RA	1.318 (1.061–1.637)	0.013	1.151 (0.602–2.200)	0.670
RV	1.636 (1.187–2.255)	0.003	0.888 (0.444–1.773)	0.736
sPAP (RHC)	1.697 (1.276–2.257)	<0.001	0.404 (0.099–1.650)	0.207
mPAP	1.832 (1.353–2.479)	<0.001	17.365 (5.757–52.381)	<0.001
dPAP	1.565 (1.157–2.116)	0.004	0.140 (0.048–0.407)	<0.001
PAWP	1.359 (1.021–1.810)	0.036	0.480 (0.251–0.916)	0.026
LVEDP	1.045 (0.719–1.519)	0.818		
Creatinine	1.083 (0.852–1.375)	0.515		
CRP	1.242 (1.014–1.521)	0.036	1.266 (0.816–1.965)	0.293
Hb	0.942 (0.691–1.285)	0.708		
BNP	1.935 (0.825–4.540)	0.129		
sST2	1.501 (1.214–1.855)	<0.001	1.914 (1.209–3.032)	0.006

## Data Availability

The data presented in this study are available on request from the corresponding author.

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
