# Peer review of "Soluble ST2 as a Potential Biomarker for Risk Assessment of Pulmonary Hypertension in Patients Undergoing TAVR?"

_life, 2022, doi:10.3390/life12030389_

Round 1

Reviewer 1 Report

I am interested in the study entitled “Soluble ST2 as a potential biomarker for detection of pulmonary hypertension in patients undergoing TAVR? ” by Elke Boxhammer et al. I think the data is interesting. I raise several points outlined below.

1) I want to know when the patients were underwent RHC, before TAVR or after TAVR? How many days or months were there, from RHC to collecting blood sample with sST2 or TAVR?

2) In this study, only 152 patients were underwent RHC among total of 260 patients who planned for TAVR. I think authors should show the reason why some patients were underwent RHC and others were not. And also I think authors should show the patients characteristics of no-mPAP data, because the number of patients with no-mPAP data was too large.

3) In table 1 authors showed p-value, I guessed the p-value was for between no-PH and PH, but I think authors should write the object for p-value. If authors show the data of no mPAP patients, it may be necessary to compare also the no-PH and no-mPAP or PH and no-mPAP groups.

4) In Figure 2B, although authors showed the data of 3 groups, No PH, prec-PH, and postc-PH, and showed the difference was existed only between non-PH and postc-PH groups, I am not sure the data were analysed using Post Hoc Tests with ANOVA? I think authors should show and comment the analysis of 3 groups with ANOVA.

5) Although authors divided the patients in 3 groups as No-PH, prec-PH, and postc-PH, many papers recently used the classification as pre-capillary PH, isolated post-capillary PH, and combined post-capillary PH. I think, if possible, it could be better to analyze the data with 4 groups, no-PH, pre-capillary PH, isolated post-capillary PH, and combined post-capillary PH in this paper.

6) Although authors underwent correlation analysis at the results 3.4, I think multivariate analysis is necessary to show that PH have correlation with elevated sST2 blood levels independently, without under the influence of other factors. At the discussion 4.1, authors commented as “double stress for heart in AS with PH”, I think this part is overstatement without multivariate analysis data.

7) Authors commented as “Pathognomonic for this is probably the pulmonary vascular remodeling, which in tern leads to a right heart strain and an associated strain-mediated myocardial release”, I want to know the elevation of blood sST2 level had correlation with RV function, for example RV size (RV area or RVEDD), RV systolic function (TAPSE and RVFAC), and TR, assessed by echocardiography.

8) Although authors showed AUROC results in Figure 3, I think, before AUROC analyses, it would be better to compare the patient characteristics of dead or alive patients, and analyze the factor correlate the prognosis. And if the level of sST2 was the independent risk factor for poor prognosis, the AUROC results have much more meanings.

9) From the data of AUROC results in results 3.5 and Figure 3, the prognostic value of sST2 seemed to be more important in non-PH patients compared with PH patients, I wonder how to interpret this data.

10) In Figure 5, although authors showed the 6 figures, I think it would be better to show all data in 1 figure.

11) From my understanding, the cut-off point of 6990.12 with the sST2 value was from the data of non-PH and PH patients of figure 4. In figure 5 authors analyzed the prognosis in combination with mPAP and the sST2 value of 6690.12. This may be real, but I wonder, for example, if authors used cut-off point of 10268.78 in PH patients, how the Kaplan-Meyer curve would be changed.

12) For my feeling, the discussion part may be too long, and in some part redundant. The data, especially the detailed number, was already presented in results part and each figures.

13) Figure 2 seems to be somewhat hard to understand at a glance. I think if the data was sorted in same categories, it would be easy to understand.

14) Although the title is “Soluble ST2 as a potential biomarker for detection of pulmonary hypertension in patients undergoing TAVR?”, and in the discussion 4.2 authors wrote as “Can sST2 as a biomarker predict (PH) or even replace RHC measurement”, I think authors should analyze the factor to predict PH and underwent multivariate analysis, and should show the results as to the sST2 level is most effective to detect PH compared with other factors such as TrPG, other echocardiographic parameters, BNP, ECG, and so on. The title seems to be misleading, I think.

15) In the discussion 4.2, authors wrote as “and thus secretion of sST2 is not directly related to criteria of pressure overload”. The patients in this paper were only severe AS patients who showed high AV Vmax, so I think it seems to be overstatement.

16) I wonder and I want to know the blood levels of sST2 were changed after TAVR? And was the level of sST2 after TAVR also correlated with PH and poor prognosis?

17) I am sorry to point out the detail. The term PAWP, not PCWP, is usually used recently. And the definition of PH is gradually changing from mPAP above 25mmHg to over 20mmHg. I think either is OK in the present, but in the future, over 20mmHg seems to be a bit better.

Author Response

We have addressed the questions raised by reviewer #1, please see the attached file with our response and the revised manuscript.

Reviewer 2 Report

Poorly legible Table 1 caption above the table
Please redraft your conclusions
Please find more recent publications 

Author Response

We have addressed the questions raised by reviewers and revised the manuscript thouroughly, please see the attached files and the revised manuscript.

Round 2

Reviewer 1 Report

Thank you very much, authors answered almost all of our comments, and I appreciate the effort. I’m sorry for adding a few points to clear.

1) From Figure 2, the sST2 level is elevated in only post-capillary PH patients, and in pre-capillary PH patients the sST2 level was not elevated, compared with no PH patients. I am just wondering the possibility that both of PH and the elevation of sST2 level was caused mainly by high PAWP, and there is no direct correlation between PH and sST2 level in this cohort. Of course, I know previous papers showed the correlation of PH and sST2 level, there is a good possibility PH and sST2 level was also correlated directly in this cohort.

I wonder how to prove direct correlation of PH and sST2 level. For example, in table 2 authors underwent multivariate analysis as to the factor to have relationship with PH, but PAWP and RAP was excluded from the analysis. I think if authors could show the data as to the sST2 level was correlated with mPAP independently with PCWP and RAP, it could be supportive as to the direct correlation of PH and sST2 level.

And also at table 3, authors showed the correlation coefficient of Spearman correlation analysis, the figure of scatter plot of sST2 level to mPAP and sST2 level to PCWP, (and also mPAP to PCWP?) could show some information.

2) I wonder as to the change of the data of correlation analysis and table 3 with rivision. The selection effect of correlation analysis could have large impact on the correlation coefficients and the p-value. I think, in this paper authors wanted to analyzed the significance of sST2 level in severe AS patients, so the data as to overall cohort or total patients who underwent RHC would be more important than the data as to selected patients of no-PH or PH.

Author Response

From Figure 2, the sST2 level is elevated in only post-capillary PH patients, and in pre-capillary PH patients the sST2 level was not elevated, compared with no PH patients. I am just wondering the possibility that both of PH and the elevation of sST2 level was caused mainly by high PAWP, and there is no direct correlation between PH and sST2 level in this cohort. Of course, I know previous papers showed the correlation of PH and sST2 level, there is a good possibility PH and sST2 level was also correlated directly in this cohort.

Answer: Thank you for this excellent point. We have given the topic of "precapillary" vs. “postcapillary“ PH and the possible association with PAWP a separate chapter in the discussion section.

I wonder how to prove direct correlation of PH and sST2 level. For example, in table 2 authors underwent multivariate analysis as to the factor to have relationship with PH, but PAWP and RAP was excluded from the analysis. I think if authors could show the data as to the sST2 level was correlated with mPAP independently with PCWP and RAP, it could be supportive as to the direct correlation of PH and sST2 level.

Answer: I actually had to smile at this question, because in this regard we had an extended discussion with the university statistician we consulted after the 1st revision. He recommended that we deliberately avoid using right heart catheterization data in the binary logistic regression. The reason for this is that the criterion to be measured PH vs non-PH is already given by the outcome (mPAP) of the RHC. If all right heart catheter data, which are related to mPAP to a circumscribed extent, were now included in the binary, logistic regression, this would lead to considerable distortions and to results that could no longer be meaningfully evaluated. I therefore followed this advice.

And also at table 3, authors showed the correlation coefficient of Spearman correlation analysis, the figure of scatter plot of sST2 level to mPAP and sST2 level to PCWP, (and also mPAP to PCWP?) could show some information.

Answer: A new Figure 3 for better visualization of sST2 and RHC data was added. The further figures were numerically adapted in the running text.

I wonder as to the change of the data of correlation analysis and table 3 with rivision. The selection effect of correlation analysis could have large impact on the correlation coefficients and the p-value. I think, in this paper authors wanted to analyzed the significance of sST2 level in severe AS patients, so the data as to overall cohort or total patients who underwent RHC would be more important than the data as to selected patients of no-PH or PH.

Answer: To prevent selection effects, the correlation table was again divided into overall cohort, PH group and no PH group (back to the roots - table division before the 1st revision). In the first revision, patients without mPAP data were analyzed separately (simultaneously to the baseline characteristic of Table 1). It is absolutely correct that no relevant conclusions can be drawn from this.  Therefore, in addition to the table, the results section was also changed accordingly.

Reviewer 2 Report

Ok 

Author Response

We thank the reviewer of his/her positive response. He have performed another revision of the manuscript, please see also answers to reviewer #1.
